# Recent Studies on Protective Effects of Walnuts against Neuroinflammation

**DOI:** 10.3390/nu14204360

**Published:** 2022-10-18

**Authors:** Bing Tan, Yuxi Wang, Xudong Zhang, Xiangjun Sun

**Affiliations:** 1Department of Food Science and Engineering, School of Agriculture and Biology, Shanghai Jiao Tong University, Shanghai 200240, China; 2Germplasm Bank of Wild Species, Kunming Institute of Botany, Chinese Academy of Sciences, Kunming 650201, China

**Keywords:** walnuts, neuroinflammation, neurodegenerative diseases, oxidative stress, antioxidant activity, gut microbes

## Abstract

Neuroinflammation plays a significant role in the aging process and the pathophysiology of neurodegenerative diseases, such as Alzheimer’s disease. Accordingly, possible therapeutic strategies aimed at anti-inflammatory effects may be beneficial to brain health. Walnut kernels contain large quantities of unsaturated fatty acids, peptides, and phenolic compounds that have anti-inflammatory effects. The long-term intake of walnuts has been found to improve cognitive function and memory in rats and humans. However, the modulatory effect of walnuts on neuroinflammation has received much less attention. This review focuses on the potential influence and main regulating mechanisms of walnuts and their active ingredients on neuroinflammation, including the regulation of microglia activation induced by amyloid β or lipopolysaccharides, inhibition of peripheral inflammation mediated by macrophages, reduction in oxidative stress by decreasing free radical levels and boosting antioxidant defenses, and control of gut microbes to maintain homeostasis. However, the majority of evidence of the beneficial effects of walnuts or their components on neuroinflammation and neurodegeneration comes from experimental work, whereas evidence from clinical studies on the beneficial effects is scarcer and less conclusive. This review aims to provide new insights into the neuroinflammation-regulating mechanisms and natural active ingredients of walnuts and the development of walnut-based functional foods for the alleviation of neurodegenerative diseases.

## 1. Introduction

Neuroinflammation, an early event in neurodegenerative diseases, is an innate immunological response of the central nervous system (CNS). Systemic inflammation may predispose the microglia and astrocytes to a proinflammatory state, which is related to neurodegenerative diseases, such as Alzheimer’s disease (AD) and mild cognitive impairment [1]. Aging is an important cause of these neurodegenerative diseases. Plaque deposits occur in the aging brain because of reduced clearance of amyloid β (Aβ) plaques [2] and other toxic substances, which triggers a neuroinflammatory response [3,4,5], further inducing microglia and astrocytes to stimulate the production of free radicals and secretion of proinflammatory cytokines [6]. The proinflammatory cytokines released by the neuroinflammatory cascade may lead to decreased cognitive function [7]. For example, elevated levels of proinflammatory cytokines, such as tumor necrosis factor-α (TNF-α), interleukin (IL)-1β, and interleukin (IL)-6 are observed in the brains of AD patients [8]. In the process of human aging, neuroinflammation in the brain induces structural and functional changes in the hippocampus, a region critical for spatial learning and memory, and is characterized by a progressive decline in learning and memory [9]. Therefore, there has been increasing interest in strategies based on anti-inflammatory interventions in the prevention and treatment of neurocognitive disorders. Because neuroinflammation plays a pivotal role in the pathogenic cascade of neurodegenerative diseases, treatment strategies for inhibiting neuroinflammation are beneficial in the prevention of neurocognitive disorders. Rational intake of food nutrients and nutraceuticals may be a more comprehensive means of improving the health of patients with neurodegenerative diseases in addition to medication. Intervention strategies for preventing AD or delaying cognitive decline are of great significance for elderly individuals. A healthy diet rich in antioxidants and anti-inflammatory phytochemicals provides one of the most effective and inexpensive methods for the early prevention of neurodegenerative diseases.

Walnuts (*Juglans regia*) have been used for thousands of years as a healthy food and folk medicine. Many studies have confirmed that walnuts have strong antioxidant and anti-inflammatory functions. Walnut kernels contain large quantities of unsaturated fatty acids, peptides, proteins, and phenolic compounds, which have anti-inflammatory properties [10,11]. The long-term intake of walnuts has been found to improve cognitive function and memory in rats and humans. In animal experiments, consuming a walnut-enriched diet for up to 14 and 19 months significantly improved the cognitive ability of elderly mice and rats, respectively [12,13]. In human clinical trials, healthy adults aged 67–75 years without any cognitive impairment were supplemented with 15 g of walnuts daily, and they showed better cognitive function and memory compared to a control group that did not receive walnuts in their daily diet [14,15]. A population-based prospective cohort study related to all types of nuts suggested that a higher long-term intake of walnuts was associated with the better cognitive performance of older women (mean age: 74 years) [16]. Furthermore, Cahoon et al. conducted a systematic review and meta-analysis suggesting that walnut intake may have a beneficial effect on cognition-related outcomes, although with a low level of confidence [17]. The large 2-year Walnuts and Healthy Aging (WAHA) randomized controlled trial of walnuts for treatment of age-related cognitive decline found no major improvement in the whole cohort but cognitive benefit in the more at-risk participants [18]. These studies indicated that walnuts could be used as an ingredient in functional foods to attenuate cognitive dysfunction. In addition, neuroinflammation plays a significant role in cognitive decline, indicating that the beneficial effects of walnuts on cognitive function may be associated with anti-inflammatory activity, which may be supported by other relevant results from the WAHA trial [19]. However, there are no definitive efficacy data or mechanisms that confirm the relationship between neuroinflammation and individual walnut components. Against this background, our review summarizes previous reports on the relationship between walnut intake and neuroinflammation attenuation and describes the role of individual walnut components in their anti-inflammatory function. A literature search was performed using Pubmed and Web of Science. The search was based on keywords such as walnuts, neuroinflammation, anti-inflammatory activity, neurodegenerative diseases, aging, Alzheimer’s disease, oxidative stress, antioxidant activity, gut microbes, unsaturated fatty acid, peptide, and phenolic compound. The search strategy included human studies, animal studies, and in vitro studies. Standard verification tools were used to assess the risk of bias and the strength of the evidence.

## 2. Studies of the Inhibitory Effects of Walnuts on Neuroinflammatory Cascades Using In Vivo and In Vitro Models

Several in vivo animal models, such as aged rats, and mice injected with lipopolysaccharide (LPS), Aβ protein, and D-galactose (D-gal), were used to evaluate memory function and detect neuroinflammation. A study found that feeding 19-month-old rats with a 6% or 9% walnut diet significantly reduced phosphorylation of nuclear factor-κB (NF-κB) in the hippocampus [20]. Intraperitoneal injection of LPS, which stimulates inflammatory responses and exerts detrimental neurobiological effects, was utilized to induce neuroinflammation and thus triggered memory deficits in a mouse model. The walnut (*J*. *regia*) protein hydrolysate (666 mg/kg) ameliorated the memory deficits induced by LPS by normalizing the inflammatory response [21]. Accumulating evidence suggests that extracellular deposition of Aβ is an important marker of the pathogenesis of neurodegenerative diseases such as AD. Numerous studies have demonstrated that the injection of Aβ into the hippocampi of mice results in subsequent deposition of Aβ and triggers a neurodegenerative cascade in the brain, leading to cognitive impairments [22,23], and for this reason, Aβ-induced mice have been considered an animal model of neurodegenerative disease. Zou et al. reported that the expression levels of proinflammatory cytokines, including IL-6, IL-1β, and TNF-α, increased significantly in Aβ_25-35_-induced AD mice, and these factors were markedly attenuated by oral gavage of walnut (*Juglans sigilata* Dode) protein hydrolysate at doses of 200, 400, or 800 mg/kg in distilled water once daily for 5 weeks [24]. Kim and colleagues also demonstrated that walnut (*Juglans regia* L.) extract regulated the expression of IL-1β, TNF-α, tumor necrosis factor receptor 1 (TNFR1), and cyclooxygenase-2 (COX-2) related to neuroinflammation in Aβ_1__-42_-induced mice [25]. Chronic exposure to D-gal, which is widely used to establish a model of accelerated aging, induced oxidative stress and inflammation, and thus resulted in neurodegeneration in mice [26]. Treatment with walnut protein hydrolysates (1 g/kg) for 90 days alleviated oxidative stress, reversed cholinergic dysfunction, and suppressed the release of TNF-α and IL-1β in the hippocampus of the D-gal-treated mice [27].

The protective effects of walnuts against neuroinflammation have also been studied in various neural cell culture models. The progression of neurodegenerative diseases frequently involves the activation of microglia. Microglia are resident macrophages of the CNS that are activated only when stimulation occurs. Additionally, microglia participate in host defense and tissue repair by secreting a variety of inflammatory cytokines such as TNF-α, IL-6, and IL-1β [28,29]. However, prolonged chronic inflammation induces sustained proliferation of microglia and continuous secretion of inflammatory cytokines, leading to neuronal damage and dysfunction associated with the inflammatory response. A number of studies confirmed that microglia in neuroglia cultures secrete proinflammatory cytokines when treated with Aβ or LPS, and thus Aβ or LPS-treated microglia cells were considered an in vitro model of microglial activation [30]. Thangthaeng et al. demonstrated that treatment with whole walnut (*J*. *regia*) extract (1%, 3%, or 6%) 1 h prior to LPS treatment was effective in preventing LPS-induced upregulation of inducible nitric oxide synthase (iNOS) expression in HAPI microglial cells [31]. The protective effects of the walnut extract on LPS-induced activation were also identified in BV-2 microglial cells. Treatment of cells with walnut extract (*J*. *regia*) prior to LPS stimulation attenuated the production of nitric oxide and expression of iNOS. The walnut extract also induced a decrease in TNF-α production. These anti-inflammatory effects of walnut were dependent on the activation of phospholipase D2 [32]. Another study further reported that serum metabolites from walnut-fed aged rats attenuated LPS-induced proinflammatory factor TNF-α, COX-2, and iNOS in BV-2 microglial cells, suggesting that walnut serum metabolites provide anti-inflammatory protection in brain cells [33]. The synergy between peripheral inflammation and central nervous inflammation deserves attention. Macrophages induced by LPS can stimulate the inflammatory response of microglia, indicating that macrophage-mediated peripheral inflammation promotes neuroinflammation. [34]. Furthermore, inflammatory microglia attracted peripheral innate immune cells by secreting various chemokines and thus aggravated neuroinflammation. The complex interactions between these innate immunity cells exacerbated the damage to the central nervous system. Therefore, neurodegenerative diseases can be alleviated by blocking signaling between peripheral cells and the central nervous system [35]. Wang et al. reported a low molecular weight peptide, leucine–proline–phenylalanine (LPF), isolated from walnut protein hydrolysates, which attenuated memory impairment by reducing neuroinflammation and oxidative stress in the brain tissue of mice induced by intraperitoneal injection of LPS, and this was also associated with the suppression of iNOS, COX-2, and TNF-α mRNA expression in macrophages [36]. The rat pheochromocytoma cell line PC12 with a morphology and physiological function similar to neurons is a well-established cell model for studying the cellular biology of neurons [37]. Hydrogen peroxide-treated PC12 cells were used to investigate oxidative stress and inflammation-associated neuronal injury [38]. A previous study reported that treatment with a walnut (*Juglans mandshurica* Maxim) protein-derived peptide EVSGPGLSPN inhibited NF-κB pathway activation, suppressing the downstream inflammatory factors IL-1β and TNF-α in H_2_O_2_-induced PC12 cells. This information indicated the potential of the walnut peptide to inhibit the neurotoxic cascade in the brain and thus prevent aging-related neurodegeneration [39].

## 3. Anti-Inflammatory Components of Walnuts

### 3.1. Polyunsaturated Fatty Acids

Walnuts are rich sources of polyunsaturated fatty acids, such as linoleic acid (LA, C18:2n-6) and alpha-linolenic acid (ALA, C18:3n-3) [40]. Accumulating evidence has shown that LA has neuroprotective properties. Tu et al. observed that microglial inflammation induced by palmitic acid treatment can be effectively reduced by LA, suggesting that LA exerts neuroprotective effects by alleviating microglia activation [41]. In a study using an AD drosophila model, LA inhibited the cytotoxicity of Aβ [42]. Consistently, LA demonstrated potential anti-inflammatory effects on Aβ-stimulated PC12 cells. The elevated TNF-α and IL-1β levels decreased; the increased production of proinflammatory mediators, including nitric oxide and prostaglandin E2 (PGE2), was inhibited; and LA produced a decrease in the expression of phosphorylated nuclear factor of kappa B (*p*-p65) and phosphorylated nuclear factor of kappa light polypeptide gene enhancer in the B-cell inhibitor (*p*-IκB) [43]. Moreover, LA was found to suppress the expression of proinflammatory cytokines including TNF-α, IL-6, and IL-1β in RAW 264.7 macrophage cells [44], further extending its potential inhibition on neuroinflammation. Conjugated linoleic acid (CLA), positional, and geometric isomers of LA [45] induce a decrease in inflammatory factors, including TNF-α and IL-1β in primary human astrocyte cultures, suggesting a potential nutritional role in regulating astrocyte inflammatory responses [46,47]. Furthermore, CLA can be integrated and metabolized into brain tissue, further extending its antineuroinflammatory effects [48]. In addition, CLA isomers (trans-10, cis-12 and cis-9 CLA, and trans-11 CLA) were observed to reduce the concentration of PGE2 and the expression of proinflammatory cytokines in macrophages obtained from blood [49]. Treatment with ALA increased glial cell viability and significantly attenuated Aβ_25−35_-induced excessive production of nitric oxide and inflammatory cytokines IL-6 and TNF-α [50]. Peripheral blood mononuclear cells from hypercholesterolemic humans supplemented with ALA secreted lower levels of proinflammatory cytokines (TNFα, IL-6, and IL-1β) compared with the control group [51]. Similarly, ALA had a strong inhibitory effect on the expression level of iNOS and the production of LPS-induced TNF-α and IL-6 by reducing the translocation of an NF-κB subunit, whereas ALA also increased the secretion of the anti-inflammatory cytokines IL-10 in RAW264.7 macrophages [52,53,54]. Moreover, in vivo studies further demonstrated that the oral administration of ALA regulated NF-κB and IL-1β in the mouse brain [55].

### 3.2. Phenolic Compounds

Walnuts contain significant quantities of phenolic compounds, of which phenolic acids and flavonoids are the main phenolic types [56]. In particular, ellagic acid, gallic acid, chlorogenic acid, catechin, and quercetin, which exhibit a high relative abundance in walnuts [57,58], have been studied for their effects on neuroinflammation [59,60]. Ellagic acid (2,3,7,8-tetrahydroxybenzopyrano [5,4,3-cde] benzopyran-5-10-dione, EA) has long been known for its antioxidant and anti-inflammatory properties [61]. Recent studies have shown that EA has a neuroprotective effect against the excessive production of proinflammatory cytokines. In a study using EA to treat an LPS-induced dopamine neuronal damage rat model, EA had a profound effect on protecting dopamine neurons via suppression of the microglial nucleotide-binding domain-like receptor protein 3 (NLRP3) inflammasome signaling activation and reduction of proinflammatory cytokine (IL-1β, TNF-α, and IL-18) expression [62]. In another study of Sprague Dawley rats fed with a high-fat diet supplemented with equimolar concentrations of EA for 13 weeks, EA exhibited good anti-inflammatory performance, for example by reducing serum cytokines IL-6, IL-1β, and TNF-α [63]. Gallic acid (3,4,5-trihydroxybenzoic acid, GA) has a wide range of antioxidant, anti-inflammatory, and antimicrobial properties [64]. GA was found to significantly inhibit PGE2 production in LPS-treated RAW 264.7 cells [65]. Furthermore, GA-treated mice exhibited reduced levels of proinflammatory cytokines and reduced infiltration of CD4^+^CD45^+^ T cells and monocytes into the central nervous system, suggesting that GA can be considered a potential inflammatory therapeutic agent [66]. Chlorogenic acid ((1S,3R,4R,5R)-3-[[3-(3,4-dihydroxyphenyl)-1-oxo-2-propen-1-yl]oxy]-1,4,5-trihydroxycyclohexanecarboxylic acid, CGA) is an ester formed from caffeic acid and L-quinic acid. It was reported that the antioxidant and anti-inflammatory properties of CGA were expected to help to alleviate cognitive and memory impairment [67,68,69]. CGA was found to have an anti-inflammatory effect on IL-1β, TNF-α, and IL-6 production [70]. It was also found to repress the activation of the NLRP3 inflammasome [71] in LPS-stimulated murine RAW 264.7 macrophages and in BV-2 microglial cells by effectively downregulating the NF-κB pathway [72]. As the important catechin in walnuts, epigallocatechin 3-gallate (EGCG) is a typical flavone-3-ol polyphenolic compound with eight hydroxyl groups, showing notable anti-inflammatory activities [73]. Bao et al. showed that oral administration of EGCG (50 mg/kg) for 4 months markedly attenuated the cognitive impairments in APP/PS1 transgenic mice used as an AD model, alleviated microglia activation and expression of the proinflammatory cytokine IL-1β, and increased the level of anti-inflammatory cytokines IL-10 and IL-13 [74]. Kawai et al. reported that EGCG downregulated CD11b expression in CD8^+^ T cells and further inhibited their infiltration into sites of inflammation [75]. Li et al. found that treatment with EGCG significantly reduced IL-1β, interferon-γ (INF-γ), and TNF-α levels in an autoimmune thyroiditis rat model through suppression of the NF-κB pathway [76]. Cheng et al. demonstrated that EGCG-loaded liposomes could reduce the production of nitric oxide and TNF-α in BV-2 microglia following LPS exposure [77]. Epicatechin (EC), another important catechin in walnut, was found to downregulate the proinflammatory cytokines IL-1β, IL-6, and TNF-α in LPS-induced RAW264.7 cells [78]. Quercetin, a representative of the flavonol family of compounds abundant in walnuts [79], was reported to reduce the expression of iNOS in LPS-activated BV-2 microglia and restrain the activation of NF-kB, indicating its potential to attenuate inflammatory diseases of the central nervous system. In addition, quercetin inhibited the production of proinflammatory cytokines, such as TNF-α, IL-1β, and IL-6, and reduced cyclooxygenase and lipoxygenase expression in mast cells [80]. Quercetin was also reported to inhibit the production of TNF-α, IL-6, and IL-1β in LPS-activated human mononuclear U937 cells [81].

### 3.3. Walnut Protein-Derived Peptides

A growing number of bioactive peptides have been identified in the protein fraction of walnuts [82]. It was found that walnut peptides decreased the neuroinflammation caused by superfluous quantities of inflammatory cytokines in neurodegenerative disorders [83]. For example, treatment with the walnut peptides LPF, GVYY, or APTLW inhibited the overproduction of proinflammatory mediators, including nitric oxide and PGE2, and reduced the expression level of TNF-α, IL-1β, and IL-6 in BV-2 microglia stimulated with LPS [21]. Another walnut peptide, WEKPPVSH, was also found to significantly mitigate the secretion of TNF-α, IL-1β, and IL-6, and downregulated the expression of iNOS, COX-2, and *p*-IkB/IkB in LPS-activated BV-2 microglia [84]. In addition, the walnut-derived peptide EVSGPGLSPN attenuated inflammatory factors IL-1β and TNF-α in H_2_O_2_-induced PC12 cells. This indicated the potential of the walnut peptide to inhibit inflammatory cascades in the brain and thus prevent aging-related neurodegeneration [39]. It was found that LPS-induced macrophages stimulated the activation of microglia, indicating that macrophage-mediated peripheral inflammation may accelerate neuroinflammation [34]. Wang et al. showed that the walnut peptide leucine–proline–phenylalanine downregulated the mRNA expression of inflammatory mediators, such as INOS, COX-2, and TNF-α, in LPS-treated RAW264.7 macrophage cells, suggesting the potential of walnut peptides against neuroinflammation through regulation of peripheral immune cells [36].

## 4. Possible Mechanisms

### 4.1. Antioxidant and Anti-Inflammatory Activity

The imbalance between reactive oxygen species (ROS) and the cellular defense antioxidant mechanisms, known as oxidative stress, is a vital factor in aging and disease. The brain is particularly vulnerable to oxidative damage because the brain tissues have high oxygen consumption but low antioxidant levels and poor regeneration. Moreover, oxidative stress is considered to induce a variety of inflammatory responses [85]. Some ROS can promote intracellular signaling cascades, upregulating the expression of related proinflammatory genes. Excessive oxidative stress and inflammatory toxicity are considered to be among the main causes of cerebral neurodegeneration [86]. Growing evidence indicates that neuroinflammation is an early event in neurodegeneration pathogenesis, which is usually related to oxidative damage [87]. Oxidative stress and inflammation also play pivotal roles in other brain disorders, such as Parkinson’s disease and several age-related chronic diseases [88,89].

Walnuts are rich in components that have antioxidant and anti-inflammatory properties that work together to inhibit inflammation and oxidative damage. The antioxidants of walnuts are mainly composed of flavonoids, phenolic acid, folate, gamma-tocopherol, selenium, juglone, proanthocyanidins, and polyunsaturated fatty acids. These antioxidant ingredients may also have highly potent anti-inflammatory effects [90,91]. A study showed that walnut extract suppressed Aβ-induced abnormalities of mitochondrial function by ameliorating ROS in mouse brain tissue. Furthermore, the expressions of neuroinflammation-associated molecules including TNF-α, TNFR1, *p*-IκB, COX-2, and IL-1β, were regulated by walnut extract [25]. Another study showed that ROS levels, lipid peroxidation, and protein oxidation in AD transgenic mice fed a diet containing 6% or 9% walnuts were significantly reduced, whereas the activities of antioxidant enzymes were significantly increased [92]. Similarly, the walnut extract was observed to regulate superoxide dismutase (SOD) and glutathione (GSH) levels in Aβ_1-42_-induced mice [24]. These findings suggested that walnuts, in addition to reducing free radical levels, enhance antioxidant defenses and reduce oxidative damage to lipids and proteins [93]. Among the active ingredients of walnuts, phenolic compounds have been shown to possess significant antioxidative and anti-inflammatory properties. Catechins were found to scavenge 1,1-diphenyl-2-picrylhydrazyl (DPPH) radicals and improve the production of BV-2 microglia-derived nitric oxide and TNF-α following LPS treatment [82]. Quercetin attenuated ROS production by regulating the heme oxygenase1/nuclear factor erythroid 2-related factor (Nrf2) pathway and inhibited the activation of the NF-kB pathway in LPS-activated BV-2 microglial cells and macrophages [80]. Ellagic acid administration downregulated abnormal ROS generation in a dose-dependent manner in the hippocampi of Wistar rats, and significantly altered inflammatory markers, including IL-1β, TNF-α, and INF-γ [61]. Gallic acid significantly increased the expression and activity of antioxidant enzymes [94], and showed a significant decrease in oxidative stress markers and inflammatory cytokines, including TNF-α, IL-1β, and IL-6 in H_2_O_2_-treated rat embryonic fibroblast cells [95]. Walnut protein-derived peptides, another important type of walnut component, were observed to ameliorate memory deficits induced by LPS, a process that is associated with the normalization of the inflammatory response in the brain of mice. ROS homeostasis might contribute to the anti-inflammatory effects, as the activities of SOD and catalase (CAT) increased significantly, and the extensive increase in the level of malondialdehyde (MDA) in the brain was reversed after treatment with walnut protein hydrolysates [21]. These results are consistent with other studies showing that supplementation with walnut peptides regulated levels of antioxidant enzymes as well as inflammatory mediators in the brain tissue of mice [24]. The walnut peptide DWMPH was found to attenuate D-gal-induced neuronal dysfunction by increasing the activities of SOD and GSH-peroxidase and suppressing the release of proinflammatory cytokines [27]. Another walnut peptide, EVSGPGLSPN, was also observed to increase SOD, GSH-peroxidase, and CAT activities in a dose-dependent manner in H_2_O_2_-induced PC12 cells, attenuating the overexpression of IL-1β and TNF-α [39]. These findings demonstrate that walnuts and their main components could be effective against neurodegenerative diseases because of their antioxidative and anti-inflammatory activities, which may exert additive or synergistic effects. In addition, the attenuation of inflammation may be associated with the increase in antioxidant enzymes, which involves the Nrf2 signaling pathways (Figure 1).

### 4.2. Gut Modulation Activity

Gut microbiota participates in brain–gut–microbe bidirectional communication through neural, immune, humoral, and endocrine connections. The pathogenic bacteria-derived LPS induces gastrointestinal inflammation and even systemic inflammation [96]. In addition, chronic intestinal inflammation may accelerate disruption of the blood–brain barrier, increasing the permeability of LPS and proinflammatory cytokines in the brain [97]. Intestinal bacteria secrete signaling factors passing through the CNS via lymphatic and systemic circulation. This information implies that a balanced gut microbiome helps to control neuroinflammation and maintain normal CNS function [98].

Walnut consumption was found to affect the composition of the human gastrointestinal microbiota and reduce microbially derived proinflammatory factors [97]. This suggested that modulation of gastrointestinal microbiota may contribute to the beneficial health effects of walnut consumption [99]. Animals consuming walnuts displayed significantly greater microbiota species diversity. Walnut consumption enriched the probiotic-type bacteria, including *Lactobacillus*, *Ruminococcaceae*, and *Roseburia* while reducing neuroinflammation-related bacteria, including *Bacteroides* and *Anaerotruncus* [100]. Furthermore, some bioactive walnut components have been reported to improve dysbiosis of gut microbiota. A study showed that walnut oil exerted anti-inflammatory effects by decreasing the expression of TNF-α in the duodenal mucosa of mice, and the relative abundance in gut microbiota shifted from more pathogenic bacteria, such as *Helicobacter*, toward the probiotic *Lactobacillus* [101]. Another study found defatted walnut powder extract also remodeled the disordered microflora in C57BL/6 mice caused by a long-term high-fat diet, producing a decrease in *Erysipelotrichia*, *Firmicutes*, and *Actinobacteria*, and an increase in *Bacteroidetes*, *Clostridiales*, *Bacteroidales* S24-7, *Prevotellaceae*, and *Bacteroides* [102]. The walnut-derived peptide leucine–proline–phenylalanine mitigated serum inflammatory cytokine levels in a dextran sulfate sodium-induced colitis mice model and reversed the dysbiosis of gut microbiota, including recovery of microbiota diversity and an increase in the relative abundance of the families *Lachnospiraceae* and *Ruminococcaceae* [103]. Furthermore, polyphenol extract from walnut meal was found to significantly limit the serum LPS, TNF-α, and IL-6 levels, and impede changes in intestinal flora in feces, mainly *Firmicutes*, *Bacteroidetes*, and *Proteobacteria* [104]. In addition, the ingestion of walnut meal dietary fiber (WMDF) effectively improved the gut microbiota disorder caused by a high fructose (HF) diet, producing higher relative abundance of *Firmicutes*, *Actinobacteria*, *Proteobacteria*, *Deferribacteres*, *Tenericutes*, and *Patescibacteria*, and a much lower relative abundance of *Bacteroidetes* and *Verrucomicrobia*. Consistently, the concentrations of IL-1β, IL-6, and TNF-α in the serum of HF-fed mice were restored by WMDF administration [105]. Peripheral inflammation has been shown to aggravate LPS and proinflammatory cytokine-induced activation of microglia and astrocytes. Walnut consumption was found to attenuate peripheral inflammation by remodeling disordered gut microflora, which may contribute to prevention of inflammation in the CNS (Figure 2). However, the correlation between changes in the gut microbial communities following a walnut diet and inflammation are complicated, and the microbial markers remain unclear, requiring further elucidation by data from epidemiological investigations and clinical studies.

## 5. Conclusions

Inflammation plays a significant role in the pathophysiology of neurodegenerative diseases. A walnut diet was found to significantly reduce the phosphorylation of NF-κB, amyloid beta, and oxidative stress in mouse hippocampi induced by LPS, and to attenuate the expression of proinflammatory cytokines, including IL-6, IL-1β, and TNF-α. These findings suggest that a nutritional diet rich in walnuts may be effective in improving chronic inflammation and neurodegeneration. Walnuts have multiple anti-inflammatory ingredients which may have additive or synergistic effects in inhibiting inflammation (Table 1). However, the level of evidence is weak given that the bulk of the studies were not clinical trials.

Oxidative stress promotes the development of inflammation. In addition to reducing free radical levels, walnuts enhanced antioxidant defenses and thus reduced systemic inflammation and neuroinflammation. In addition, LPS and proinflammatory cytokines induced by gut microbiota were shown to transfer via lymphatic and systemic circulation throughout the CNS, and walnut consumption attenuated this process by remodeling disordered microflora. In order to clearly understand the inhibitory effects of walnuts on neuroinflammation and the underlying mechanisms, further studies are required to investigate how walnut products regulate signals and cytokines to inhibit neuroinflammation in neurodegenerative diseases. In addition, more clinical trials are needed to determine whether the neuroinflammatory effects translate to humans.

## Figures and Tables

**Figure 1 nutrients-14-04360-f001:**
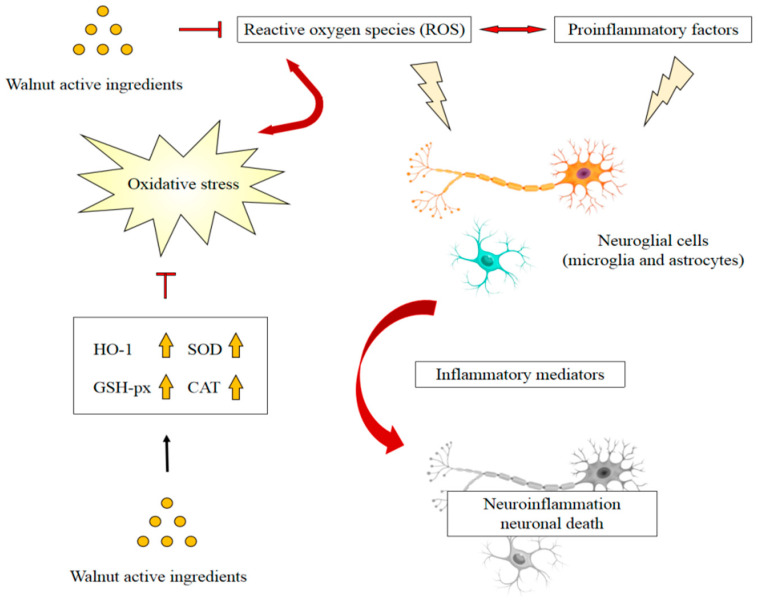
Possible mechanism of walnut active ingredients attenuating neuroinflammation by reducing oxidative stress. Adapted from [87].

**Figure 2 nutrients-14-04360-f002:**
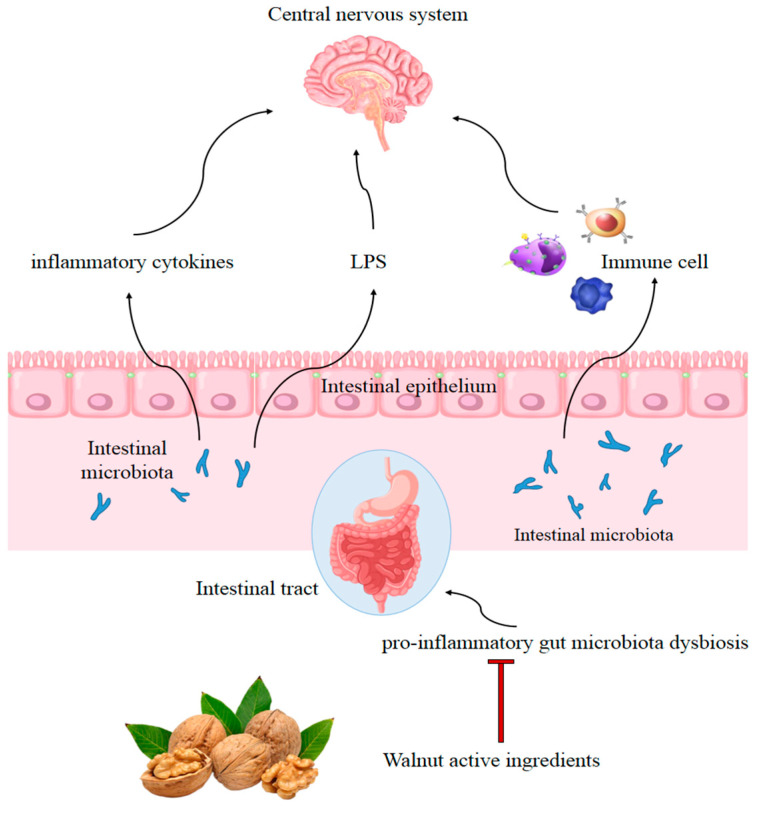
Possible mechanism of walnut consumption preventing inflammation in the CNS by remodeling disordered gut microflora. Adapted from [106].

**Table 1 nutrients-14-04360-t001:** Effect of walnut ingredients on inflammatory responses of neuronal and peripheral immune cells.

Types of Active Ingredient		Model	Dose	Results	Ref.
Fatty acids	Linoleic acid	Aβ_25−35_-treated PC12 cells	10, 50 or 100 μM	Decreased the Aβ_25-35_-elevated TNF-α and IL-1β levels by 50%; inhibited increased NO production by reducing iNOS; inhibited PGE2 by decreasing COX-2; decreased the level of *p*-p65 and *p*-IκB.	[43]
	Linoleic acid trans-10, cis-12 CLA cis-9, trans-11 CLA	Human macrophages	20 or 40 μM	Reduced PGE2 concentration by 23%; reduced COX-2 activity. Reduced PGE2 concentration by 39%; reduced the quantity of the active p65 NF-κB subunit by 55%. Reduced PGE2 concentration by 32%; reduced the quantity of the active p65 NF-κB subunit by 58%.	[49]
	Alpha-linolenic acid	LPS-stimulated RAW 264.7 cells	5, 10, 20 or 40 μg/mL	Inhibited translocation of the NF-κB subunit; downregulated inflammatory iNOS, COX-2, and TNF-α gene expression in a dose-dependent manner.	[52]
		LPS-stimulated RAW 264.7 cells	50 μM	Decreased expression levels of TNF-α and IL-6; increased the secretion of the anti-inflammatory cytokines IL-10.	[53]
		Carrageenan-induced hind paw edema in SD rats LPS-stimulated RAW 264.7 cells	5 or 10 mg/kg	Reduced rat paw edema; inhibited the accumulation of nitrite and PGE2. Inhibited the protein and mRNA expression levels of iNOS and COX-2 enzymes in a dose-dependent manner.	[54]
Phenolic acids	Ellagic acid	Arsenic-treated rats	10–20 mg/kg by mouth, in drinking water for 8–11 days	Decreased levels of mRNA and proteins TNF-α, IL-1β, and INF-γ in the hippocampus.	[61]
		LPS-elicited DA neuronal loss in SD rats LPS-stimulated BV-2 cells	50 mg/kg (oral) 1 μM	Suppressed LPS-induced activation of NLRP3 inflammasome signaling and IL-1β, TNF-α, and IL-18 protein expressions in the rat brain. Inhibited LPS-induced activation of microglial NLRP3 inflammasome signaling; eliminated production of TNF-α, IL-1β, and IL-18 in the culture medium.	[62]
		Macrophage migration inhibitory factor (MIF)-treated human peripheral blood mononuclear cells	50 μM	Inhibited MIF-mediated nuclear translocation of NF-κB.	[107]
		LPS-stimulated RAW 264.7 cells	6.25 μM 25 μM	Inhibited LPS-stimulated TNF-α. Inhibited LPS-stimulated IL-6 and PGE2 production.	[65]
	Gallic acid	LPS-stimulated RAW 264.7 cells	6.25 μM	Inhibited LPS-stimulated PGE2 production.	[65]
		MOG _35__-__55_-immunized C57BL/6 mice	2 mg/day for 10 days, injected intraperitoneally	Reduced infiltration of CD4^+^CD45^+^T cells and monocytes into the central nervous system.	[66]
		Phorbol 12-myristate 13-acetate (PMA) + calcium ionophore A23187-stimulated human mast cells (HMC-1)	1–10 µM for 2–4 h	Inhibited TNF-α and IL-6 gene expression, degradation of IκBα, and nuclear translocation of p65 NF-κB induced by PMA with A23187.	[108]
	Chlorogenic acid	LPS-stimulated RAW 264.7 cells	2–20 µM for 24 h	Attenuated NO, IL-1β, TNF-α, IL-6, cyclooxygenase-2, and NF-κB expression.	[70]
		Mongolian gerbil model of transient forebrain ischemia	30 mg/kg	Attenuated IL-2 and IL-4 protein expressions in pyramidal neurons.	[69]
Flavonoids	EGCG	Isolated peripheral blood mononuclear cells and CD8^+^T cells	25–100 µM	Inhibited infiltration of CD8^+^T cells into the sites of inflammation.	[75]
		Autoimmune thyroiditis rat model	0.5 mg/kg, three times at a 1 h interval for 3 h, injected intraperitoneally	Reduced IL-1β, INF-γ, and TNF-α levels in thyroid tissue through suppression of the NF-κB pathway.	[76]
		Rat model of cerebral ischemia/reperfusion injury	50 mg/kg, intraperitoneal injection	Inhibited cerebral ischemia/reperfusion injury by ameliorating inflammation-related molecules TNF-α, IL-1β, IL-6, NF-κB/p65, COX-2, and iNOS in the cerebellum.	[109]
	Quercetin	Human mast cells HMC-1	10 μM	Inhibited mast cell tryptase and IL-6 release.	[80]
		LPS-stimulated U937 macrophages	30 μM	Reduced the levels of TNF-α, IL-6, and IL-1.	[81]
		LPS-stimulated RAW 264.7 cells	12.5 μM	Inhibited LPS-stimulated IL-6 and PGE2 production.	[65]
Peptides	Hydrolysate (<3 kDa) Viscozyme L + pancreatin	LPS-treated mice	666 mg/kg for 21 days	Reduced NO content, normalized the overproduction of IL-6, IL-1β, and TNF-α in the brain.	[21]
	Hydrolysate	Aβ_25−35_-injected mice	400 or 800 mg/kg for 5 weeks	Decreased the levels of NO, iNOS, NF-κB p65, TNF-α, IL-1β, and IL-6 in the hippocampus.	[24]
	Hydrolysate (<1 kDa) pepsin + pancreatin	D-gal + AlCl_3_-treated mice	1 g/kg for 90 days	Suppressed the expression of TNF-α and IL-1β in the hippocampus.	[27]
	LPF	LPS-stimulated RAW264.7 cells	250, 500, or 1000 μg/mL for 24 h or 48 h	Suppressed the mRNA expression of iNOS, COX-2, and TNF-α.	[36]
	LPF, GVYY, APTLW	LPS-stimulated BV-2 cells	0.10 mM	Inhibited the overproduction of proinflammatory mediators (NO and PGE2); reduced the expression level of TNF-α,IL-1β, and IL-6.	[21]
	WEKPPVSH	LPS-stimulated BV-2 cells	25 or 50 mM	Mitigated the secretion of TNF-α, IL-1β, and IL-6; downregulated the expression of iNOS, COX-2, and *p*-IkB/IkB.	[84]
	EVSGPGLSPN	H_2_O_2_-treated PC12 cells	100 μM	Suppressed the expression of IKKβ and p65 to inhibit NF-κB pathway activation; attenuated the neurotoxic cascade by overexpression of IL-1β and TNF-α.	[39]

Aβ, amyloid β; TNF-α, tumor necrosis factor-α; IL, interleukin; iNOS, inducible nitric oxide synthase; PGE2, prostaglandin E2; COX-2, cyclooxygenase-2; *p*-p65, phosphorylated nuclear factor of kappa B; *p*-IκB, B-cell inhibitor; CLA, conjugated linoleic acid; NF-κB, nuclear factor-κB; LPS, lipopolysaccharide; INF-γ, interferon-γ; NLRP3, nucleotide-binding domain-like receptor protein 3; MIF, migration inhibitory factor; PMA, phorbol 12-myristate 13-acetate; HMC-1, human mast cells; EGCG, epigallocatechin 3-gallate; LPF, leucine–proline–phenylalanine.

## Data Availability

Not applicable.

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
