# Peer review of "Recent Studies on Protective Effects of Walnuts against Neuroinflammation"

_nutrients, 2022, doi:10.3390/nu14204360_

Round 1

Reviewer 1 Report

Timely and well-written review. The figures and table are very informative and well done.

However, no methodology is provided with regards to how the search was conducted. Without this information, the risk of author bias in selecting targeted articles to support a specific view is introduced. Recommend adding information about search strategy, article review process employed, e.g., PRISMA, and risk of bias assessment tool utilized. An advantage of this is that it helps the reader discern in vivo vs. animal vs. human studies—the results of each of these studies have different strengths with regard to research findings.

Line 64: Add “did.”  “…group that did not receive…”

Lines 227-228: Is the font a different size?

Line 434: Suggest adding something like this: However, the level of evidence is weak given that the bulk of the studies were not clinical trials.

Line 444: Final sentence needs a period. Also add sentence about need for more clinical trials to determine if neuroinflammatory effects translate to humans. You might also add the value of identifying a definitive biomarker for evaluating neuroinflammation in humans.

Reviewer 2 Report

Dear authors,

after a really careful revision. I consider the review has to be rejected. Although it is interesting, the review is not very well written and has substantial flaws in the references. the figures and schemes are really poor. At the same time, there is only one table. On the other hand, sometimes it is really repetitive. Also, while the subtitles are really clear the information added in each subsection is sometimes misunderstandable.

Reviewer 3 Report

This paper contributes novel nutritional concepts with regard to the experimental evidence of benefit of walnut components on neurodegeneration. Some references are erroneous or outdated, and this needs to be corrected, as pointed out below. An important point is that the majority of evidence of a beneficial effect of walnuts or their components on neuro-inflammation and neurodegeneration comes from experimental work, while the evidence from clinical studies is both less abundant and less clear on any beneficial effect. This needs to be clearly underlined in the abstract and the conclusions. Authors should also pay attention to the points outlined below, listed in order of appearance, the resolution of which can strengthen the message of the manuscript.

1. Lines 43-44. The sentence here would read better thus: “Therefore, strategies based on anti-inflammatory interventions have been of increasing interest in the prevention and treatment of neurocognitive disorders”.

2. Line 54. You should indicate the scientific name of English walnuts (Juglans regia) or black walnuts (Juglans nigra) the first time they are named. Subsequently, this is unnecessary unless a specific walnut species is referred to (then you can use J. regia o J. nigra).

3. Line 55. Ref. 10 is inappropriate as it refers to Brazil nuts, not walnuts.

4. Line 58. Instead of refs. 11 and 12, cite here ref. 56 (and renumber it) and new ref: “Sala-Vila A, Fleming J, Kris-Etherton P, Ros E. Impact of alpha-linolenic acid, the vegetable omega-3 fatty acid, on cardiovascular disease and cognition. Adv Nutr. 2022 Feb 16:nmac016. doi: 10.1093/advances/nmac016. Epub ahead of print.

5. Lines 62-64. The daily dose of walnuts was 15 g, not 15 mg, and they were consumed together with 15 g of a mix of almonds and hazelnuts for several years. Ref. 16 is incomplete.

6. Lines 64-66. It is stated “Another clinical trial demonstrated that higher long-term intake of walnuts was associated with better cognitive performance of older women (mean age: 74 years)”. Must provide a reference!

7. In this context, authors must cite the large, 2-year WAHA randomized controlled trial of walnuts for age-related cognitive decline and summarize the results (no major improvement in the whole cohort but cognitive benefit in participants more at risk).

Sala-Vila A, et al. Effect of a 2-year diet intervention with walnuts on cognitive decline. The Walnuts And Healthy Aging (WAHA) study: a randomized controlled trial. Am J Clin Nutr. 2020;111(3):590-600. doi: 10.1093/ajcn/nqz328.

8. Lines 72-73. That the beneficial effects of walnuts on cognitive function may be associated with anti-inflammatory activity may be supported by other relevant results from the WAHA trial: Cofán M, et al. Effects of 2-year walnut-supplemented diet on inflammatory biomarkers. J Am Coll Cardiol. 2020;76(19):2282-2284. doi: 10.1016/j.jacc.2020.07.071. 

9. Lines 154-156. Refs. 38 and 39 have little bearing on the topic being discussed (?). Better use the following comprehensive review on fatty acids, with a good discussion of linoleic acid: “Wang DD, Hu FB. Dietary fat and risk of cardiovascular disease: recent controversies and advances. Annu Rev Nutr. 2017;37:423-446. doi: 10.1146/annurev-nutr-071816-064614” and again the cited recent ALA review in Adv Nutr. 2022.

10. Line 186. Omit old ref. 49 on ALA and use instead the cited ALA review in Adv Nutr. 2022.

11. Section 3.2. Phenolic compounds. Phenolic in walnuts should be discussed in reference to their relative abundance in walnuts - please see and cite: Vu DC, Vo PH, Coggeshall MV, Lin CH. Identification and Characterization of Phenolic Compounds in Black Walnut Kernels. J Agric Food Chem. 2018;66(17):4503-4511. doi: 10.1021/acs.jafc.8b01181. for comparison of individual phenolic contents in black and English walnuts.

The section 3.2 is too long. Only relevant phenolics found in walnuts and abundant in them should be succinctly discussed. Must cut wording and references by at least 50%.

12. Lines 279-280. Better write “Increasing numbers of bioactive peptides from the protein fraction of walnuts have been characterized [84]”.

13. Line 397. Table 1. There are many abbreviations throughout the Table. WITHOUT EXCEPTION, ALL ABBREVIATIONS MUST BE DESCRIBED AT THE FOOT OF THE TABLE IN ORDER OF APPEARENCE.

14. Lines 308-309. The subsection 4.1. Antioxidant activity should be renamed “Antioxidant and anti-inflammatory activity”.

15. Line 333. Ref. 12 inappropriate, as already noted (see point #4). Ref. 92 has an error in the journal (Springer Netherlands is the editor, not the journal).

16. Lines 388-389. Ref. 99 must be erroneous, no relationship with topic under discussion.

Round 2

Reviewer 2 Report

The manuscript is not modified enough to be accepted.

There are some more concerns such as English quality, images, tables and redundant information.

So, in my opinion the manuscript does not accomplish with the Nutrients standard qualities.

Author Response

Thanks for the reviewer’s comment. The manuscript has been revised according to reviewer’s comments. 

1. Wehave submited the manscript to https://www.mdpi.com/authors/english for English editing services.

2. We have redrawed Figure.1 and Figure.2 to improve the quality of images.

3. Carefully considering your comments on the redundant information, we have checked whole manuscript, and the belowsetences have been deleted:

    “have potential anti-inflammatory activities on cardiovascular disease [11,41].” (line 167 in revised version)

    “inhibit nitric oxide production and” (line 178 in revised version)

    “were shown to modulate the immune response, such as decreasing chronic inflammation and regulating inflammatory modulators” (line 181 in revised version)

    “Phenolic compounds are important nutritional ingredients in walnuts due to their excellent antioxidant and anti-inflammatory activity [10]. In particular, human experimental and epidemiological studies have shown that intake of polyphenols may be beneficial to cognition..Recent data from randomized placebo-controlled trials suggest that phenolic compounds are effective in improving neurological disorders, cerebral hypoperfusion and neuroinflammation, as well as memory and cognitive performance in older adults [57]” (line 199 in revised version)

    “Another study showed that CGA remarkably repressed the LPS-induced activation of the NLRP3 inflammasome in RAW 264.7 murine macrophages [74].” “Catechins, with two benzene rings, are characterized by di- or tri-hydroxyl group substitution of the B ring and meta-5,7-dihydroxy substitution of the A ring.”(line 226 in revised version)

    “is structurally composed of two aromatic rings and one oxygen-containing heterocyclic ring” (line 240 in revised version)

    “which was closely correlated with inhibition of NF-κB p50/P65 subunit activation ” (line 241in revised version)

    “2-(3,4-dihydroxyphenyl)-3,5,7-trihydroxy-4H-chromen-4-one”(line 241in revised version)

    “which are characterized by a 3-hydroxyflavone backbone” (line 242 in revised version)

    “glial cells, peripheral innate immune cells such as mast cells, mononuclear cells, and macrophages are also involved in neuroinflammation, secreting pro-inflammatory cytokines transferred throughout the CNS via the lymphatic and systemic circulation.” (line 245 in revised version)

    “Oxidative stress and inflammation are linked and affect one another. Oxidative stress promotes the progress of inflammatory response. Inflammatory cells secrete ROS that induce oxidative stress.”  (line 281 in revised version)

Reviewer 3 Report

There are still two small concerns with this manuscript that need to be corrected:

1. Lines 70-72. There is an error as Ref. 16 concerns a population-based prospective cohort study, not an RCT. The findings relate to total nuts, and there was only a suggestion that increased consumption of walnuts per se had a beneficial effect on cognition

2. Lines 75-76. It is written “A large 2-year WAHA randomized controlled trial of walnuts to treat age-related cognitive…” The acronym WAHA must be spelled the first time it is used, Hence the writing should be “The large 2-year Walnuts And Healthy Aging (WAHA) randomized controlled trial of walnuts to treat age-related cognitive….”

Author Response

Thanks for the reviewer’s comment. The manuscript has been revised according to reviewer’s comments. 

1. Lines 70-72. There is an error as Ref. 16 concerns a population-based prospective cohort study, not an RCT. The findings relate to total nuts, and there was only a suggestion that increased consumption of walnuts per se had a beneficial effect on cognition

Reponse: The information has been corrected (line 66-67 in revised version).

2. Lines 75-76. It is written “A large 2-year WAHA randomized controlled trial of walnuts to treat age-related cognitive…” The acronym WAHA must be spelled the first time it is used, Hence the writing should be “The large 2-year Walnuts And Healthy Aging (WAHA) randomized controlled trial of walnuts to treat age-related cognitive….”

Reponse: The sentence has been corrected (line 72-73 in revised version)